# Multi-Objective Optimization of Magnetorheological Mount Considering Optimal Damping Force and Maximum Adjustable Coefficient

**Jianghua Fu [1], Chao Huang [1,\*] , Ruizhi Shu [2], Xing-Quan Li [3,4], Ming Chen [5], Zheming Chen [1] and Bao Chen [1]**

[1]  Key Laboratory of Advanced Manufacturing Technology for Automobile Parts, Ministry of Education, Chongqing University of Technology, Chongqing 400054, China
[2]  Department of Mechanical Engineering, Chongqing University of Technology, Chongqing 400054, China
[3]  Changan Auto Global R&D Center, Chongqing 401120, China
[4]  State Key Laboratory of Vehicle NVH and Safety Technology, Chongqing 401120, China
[5]  Chongqing Ceprei Industrial Technology Research Institute, Chongqing 401332, China
[\*]  Correspondence: chaochao26@stu.cqut.edu.cn

**Abstract:** To address the problem of multiple working conditions and complex requirements in magnetorheological fluid (MRF) mounts, a high-precision damping characteristic optimization method is explored. Based on the parallel plate model, the equation of fluid motion in the inertial channel was established according to the Navier–Stokes equation, and the MRF mount damping characteristics were analyzed. Considering the fluid model to be suitable in the steady-state, the model was experimentally verified, and the extended equation was fitted. Multi-objective optimization design was carried out by considering the large damping force and adjustable coefficient as the optimization goal and external geometric dimensions as variables. According to results, under the radial-channel MRF mount structure, the magnet core depth has the least influence on the damping force; furthermore, the damping performance can be quickly improved by changing the height of the inertial channel. The addition of the extended equations further improved the accuracy of the fluid model. The multi-objective optimization design can improve the strength and uniformity of the flux density of the MRF mount damping gap. After optimization, the damping force is increased by 44.64%; moreover, when the current is increased from 1.5 to 1.8 A, the controllable force increases by only 2.26%, and the damping performance is fully exerted.

**Keywords:** magnetorheological fluid mount; optimize; damping performance; fluid motion analysis

## 1. Introduction

Magnetorheological fluid is a new type of intelligent fluid. The most important and obvious feature of MRF (magnetorheological fluid) is their reversible transition from liquid to semi-solid state in milliseconds owing to the influence of magnetic fields, and vice versa. This change in the state and properties is referred to as the magnetorheological effect [1,2]. The damping characteristics of the MRF steadily vary with the external magnetic field, which makes it an excellent working fluid for mechanical damping and isolation devices [3]. Shock absorbers with MRF as the working fluid have the advantages of low energy consumption, fast response, simple structure, adjustable damping force [4–7], and a wide range of applications in the automotive industry [8–10].

Magnetorheological mounts have been extensively studied for many years. Bai et al. and Lin et al. derived pseudohomeostatic mathematical models of MRF for the theoretical analysis of the dynamic stiffness and equivalent damping characteristics of MRF mounts [11,12]. Most optimization studies have been conducted using similar models. Aiming at the problem of automatic starting/stopping of the engine, which leads to driver discomfort, Chung et al. designed and optimized a magnetorheological mount with a large

damping force and force ratio [13]. Certain results have been obtained in engineering applications. Zheng et al. established a lumped parameter model of an MR (magnetorheological) engine mount in a single-degree-of-freedom system and carried out a multi-objective optimization design [14]. It has a certain reference value for optimization and provides a relatively standard optimization method for future research. Deng et al. conducted various studies on magnetorheological mount optimization. A magnetorheological mount for controlling the vibration and torque excitation of the engine when the vehicle is in start–stop mode was proposed, and the dynamic performance test of the magnetorheological mount unit and the vibration isolation performance test of the entire vehicle in the start–stop mode were verified [15]. To improve the quality of vehicle NVH (Noise, Vibration, Harshness), an optimization study was conducted using Isigt, MATLAB, and ANSYS software to develop an optimal co-simulation platform [16]. Aiming at the problem of premature saturation of the magnetic induction intensity when the inner diameter of the core is large, a tapered channel is proposed, and the magnetic circuit is optimized [17]. A multi-objective optimization scheme for an MR damper based on the vehicle dynamics model was proposed by introducing the damping force into the vehicle dynamics model [18]. After years of research, Deng et al. have made breakthroughs and advances in the field of magnetorheological mount optimization. However, there are still some situations, such as the difference between the theoretical derivation model used and the actual working conditions at different frequencies, that have not been taken into account.

MRF mounts have many similarities with MRF dampers, and many studies can also be used as references. Considering the improvement in the performance of MRF dampers, many methods have been proposed to improve the internal structure of the MRF dampers [19–21]. The performance of an MRF damper heavily depends on activating the magnetic circuit; therefore, the performance of an MR damper can be optimized by activating the design of the magnetic circuit [22]. Parlak et al. conducted a multi-objective optimization study with the damping force and maximum magnetic flux density as the optimization goals [23], wherein the maximum dynamic range and maximum damping force were used as the optimization goals [24]. Jiang et al. optimized the structure of an MRF damper using the NSGA-II algorithm based on the maximum dynamic range and minimum number of turns of the electromagnetic coil [25]. Hu et al. studied the effect of different piston shapes on the MRF damper performance using finite element analysis [26]. Ferdaus et al. conducted finite element analysis to study the performance variation of dampers in different configurations [27]. Dong et al. studied the function of the damping force, dynamic range, response time, and damping gap magnetic flux density consistency as the optimization goal [28]. Scholars at home and abroad have conducted considerable research on the optimal design of MRF dampers and proposed a variety of different ideas and optimization algorithms; optimized design and experimental design based on parameter programming are the two methods commonly used at present.

Most studies have not considered the difference between the theoretical models and actual work on mounts. Particularly, the theoretical model of the fluid is significantly different from the actual magnetorheological suspension at different frequencies, and the performance of the MR fluid cannot be exerted with an increase in frequency. Using the method presented in this paper, resilience at different frequencies can be estimated from the experimental data. The accuracy of the model can be further improved by comparing its reliability. Owing to the research time, the refined optimization of the proposed model is still at a relatively basic stage; however, it still provides some ideas and has a certain reference value for subsequent research.

In this study, an MRF mount with a toroidal radial channel was designed, which can have a large damping force and controllable range, even under certain geometric dimensional constraints [29]. To be more in line with engineering practice and practical application scenarios, under the premise of ensuring accuracy, the magnetic circuit model was simplified as much as possible, and the calculation speed was improved. Considering the engine vibration isolation to be the most important function, the damping force size and

controllable range are set as the optimization goals, the best optimization effect is expected, and the optimal solution of the MRF mount is obtained in combination with finite element analysis and optimization design tools.

## 2. Materials and Methods

### 2.1. Magnetorheological Mount Structure and Working Principle

A radial-channel MRF mount structure is proposed, as shown in Figures 1 and 2. The MRF mount consists of a rubber main spring, rubber base film, and core assembly. The core assembly is an axisymmetric structure, which is divided into upper and lower liquid chambers. The electromagnetic coil is wound on the magnetic core seat, the aluminum alloy partition is used to prevent magnetic leakage, and the magnetic conductive material is DT-4C material of China national standard GB/T 6983-2008.

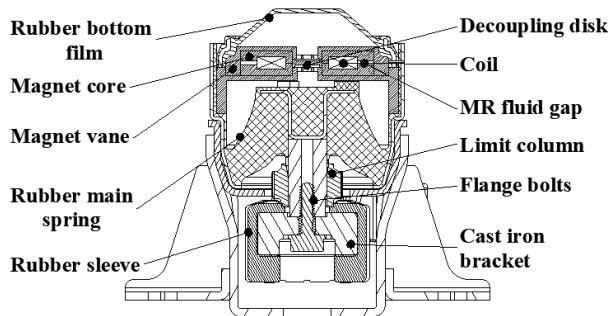

**Figure 1.** 2D structural diagram of MRF (magnetorheological fluid) mount.

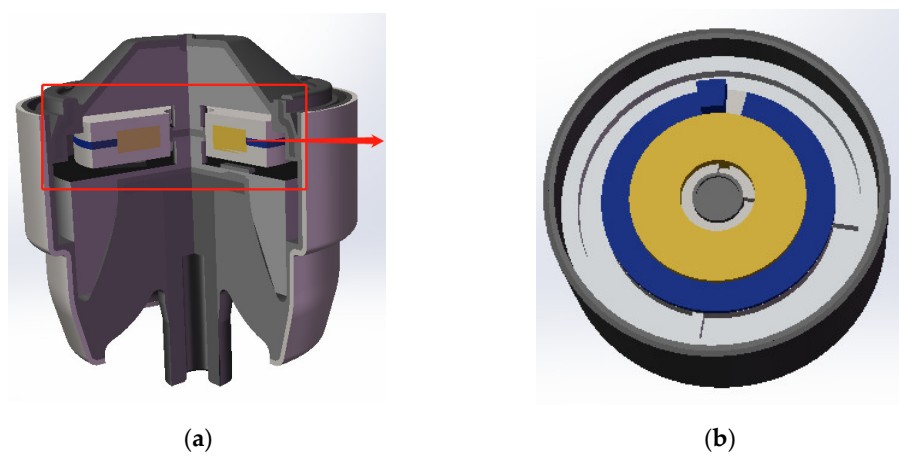

(**a**)　　　　　　　　　　　　　　　　　　(**b**)

**Figure 2.** Simplified 3D model of MRF mount: (**a**) overall cross-sectional view; (**b**) schematic diagram of the damping channel.

### 2.2. Magnetic Circuit Structure and Model Simplification

In working conditions, the engine drives the connecting rod to vibrate, and the connecting rod forces the MRF to flow between the upper and lower cavities through vibration. The controlled MRF mount produces different output damping forces by varying the current applied to the coil. The magnetic circuit is axisymmetric, and its structure can be simplified into a two-dimensional axisymmetric model, and the simplified model is shown in Figure 3.

The MRF materials used in this study were provided by the Chongqing Institute of Materials Research [30,31]. The specific parameters are listed in Table 1.

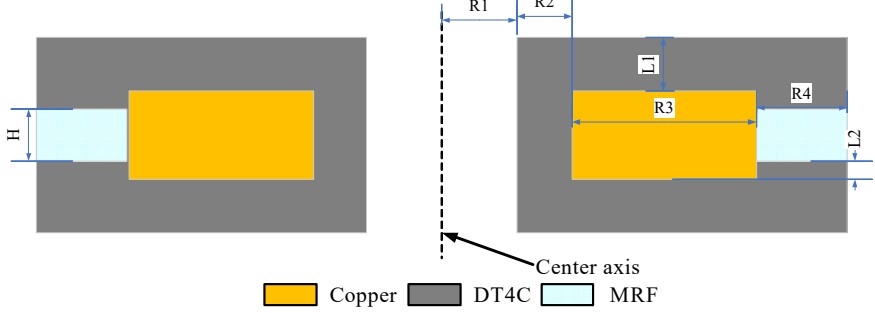

**Figure 3.** Magnetic circuit model of an MRF mount.

**Table 1.** MRF (MRF-J25T) material performance parameters.

| Parameter | Value | Parameter | Value |
|---|---|---|---|
| Fe (%vol) | 25 | Operating temperature (°C) | 40~130 |
| Carrier fluid | Hydrocarbon oils | Zero magnetic field viscosity at room temperature (Pa·s, $\gamma$ = 51/s) | ≤1.0 |
| Density (room temperature) (g/cm) | 2.65 | Shear stress (kPa, *B* = 0.5T) | ≥40 |

Because the B-H curves of DT-4C materials and magnetorheological fluids are non-linear, the nonlinear permeability of the materials was considered in the finite element analysis to improve the calculation accuracy of the magnetic field characteristics. The magnetization curves of the materials used are shown in Figures 4 and 5.

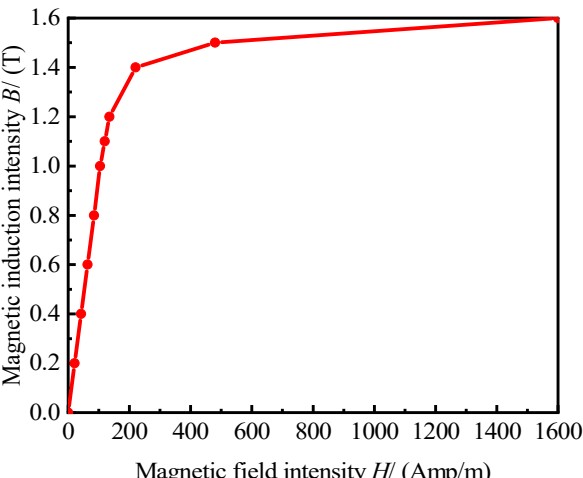

**Figure 4.** Magnetization curves of DT-4C materials.

According to the performance parameters of the MRF, the relationship between the magnetic induction intensity *B* and the shear yield stress before extrusion can be obtained using the following fitted equation:

$$\tau_{BZ} = 1.09 - 58.37B + 878.68B^2 - 987.21B^3 \tag{1}$$

where *B* is the magnetic induction intensity and $\tau_{BZ}$ is the shear yield stress.

The theoretical damping force and Navier–Stokes equation of fluid dynamics was used to calculate the theoretical damping force [32]:

$$v(z) = \begin{cases} -\frac{\Delta P_m}{\eta_0 l}\left(\frac{1}{2}z^2 - z_1 z\right), & z \in [0,\ z_1) \\ \frac{\Delta P_m}{2\eta_0 l}y^2, & z \in [z_1,\ z_2] \\ -\frac{\Delta P_m}{\eta_0 l}\left[\frac{1}{2}\left(z^2 - h^2\right) - z_2(z-h)\right], & z \in (z_2,\ h] \end{cases} \tag{2}$$

$$Q_m = \int R v(z)dz = \sum_{i=1}^{3}\int R v_i(z)dz \tag{3}$$

$$F_c = \Delta P_m A_p = F_\eta + F_\tau \tag{4}$$

where $R = R_4$ is the length of the cross-section, $h = H$ is the width of the cross-section, $Q_m$ is the flow through the damping channel, $\Delta P_m$ is the pressure drop in the controllable channel, $F_c$ is the combined force of the damping force, $F_\eta$ is the viscous damping force, and $F_\tau$ is the coulomb damping force.

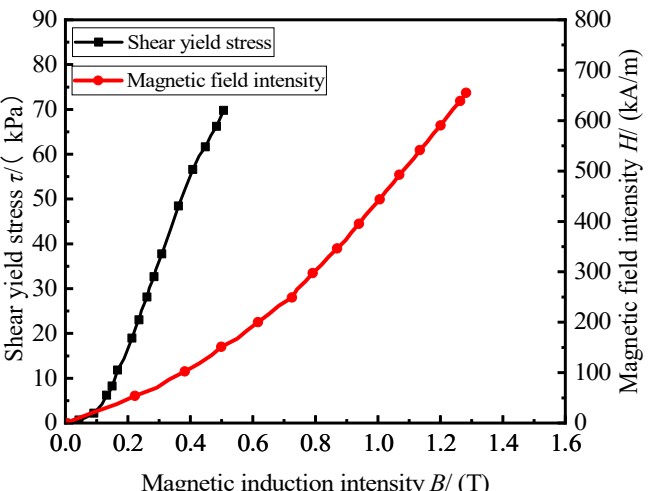

**Figure 5.** MRF (MRF-J25T) material characteristic curve.

Due to the small length of the channel, the pressure is assumed to reduce at an equal rate along the length of the controllable channel. Liquids are incompressible and symmetrical. The force balance analysis of the plunger domain is as follows:

$$z_1 + z_2 = h \tag{5}$$

$$\Delta P_m(z_2 - z_1) = 2\tau_{BZ}l \tag{6}$$

where $l$ is the channel length $l = \pi(2R_1 + 2R_2 + 2R_3 + R_4)$.

Combined with the relevant boundary conditions and mount structure, the controllable channel pressure drop and damping force can be obtained by substituting the above formula:

$$\Delta P_m = \frac{12\eta l}{Rh^3}Q_m + \frac{3l}{h}\tau_{BZ} \tag{7}$$

$$F_c = F_\eta + F_\tau = \left(\frac{12\eta l}{Rh^3}Q_m + \frac{3l}{h}\tau_{BZ}\right)A_p \tag{8}$$

where $\tau_{BZ}$ denotes the shear yield stress, $\eta$ denotes the apparent viscosity, and $A_p$ denotes the equivalent piston area.

Under the action of a magnetic field, the apparent viscosity of magnetorheological fluids can be expressed as:

$$\eta = \eta_0 + \tau_{BZ} \left/ \frac{dv_y}{dz} \right. \tag{9}$$

where $\eta_0$ is the zero magnetic field viscosity.

When the MRF mount was working, the rubber main spring was deformed by pressure. The oppressive magnetorheological fluid flows from the upper chamber to the lower chamber to pump the liquid, and the deformation of the liquid when pumping is shown in Figure 6. The blue dotted line is the shape of the rubber main spring before pumping, and the red dotted line is the shape of the main spring when pumping; thus, the green area denotes the pump area.

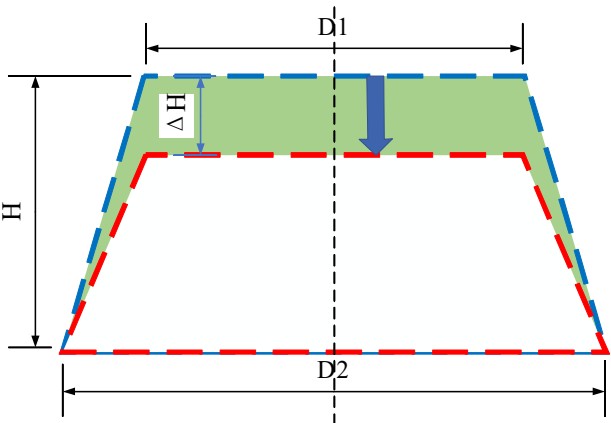

**Figure 6.** Schematic of the deformation of the rubber main spring.

The rubber main spring is not a regular shape; therefore, it is difficult to directly measure its equivalent pumping area. It can be estimated from the structural parameters, and the equivalent piston area can be calculated as follows:

$$A_{\mathrm{p}} = \frac{\pi}{12}\left(D_1^2 + D_2^2 + D_1 D_2\right) \tag{10}$$

*2.3. Magnetic Field Finite Element Simulation*

In the previous calculation, Equation (11) and China national standard GB/T 6109.1-2008 are used for simple estimation. The number of turns of the coil is set to 240 turns. The nominal diameter of the conductor is 0.500 mm. The maximum outer diameter is 0.566 mm. The breakdown voltage is 4600 V.

$$J = NI/S_e \tag{11}$$

where $J$ is the energized current density, the allowable range is $J$ = 5~12 A/mm$^2$ [33], the number of turns of the coil is $N = 240$, the current density is $J = 6$ A/mm$^2$, the effective cross-sectional area of the coil is $S_e \approx 80$ mm$^2$, and the maximum allowable current is $I_{\max} = 2$ A .

Based on the simplified model shown in Figure 3, a two-dimensional axisymmetric finite element model was established using the Maxwell software. The calculation results of the magnetic circuit analysis were substituted into the damping force calculation formula and used in the subsequent optimization process. The 2D model has a simple and regular structure, it only needs to be automatically meshed, and the mesh size is 0.5 mm. The governing equations is the equation of a static magnetic field. The magnetic circuit structure material magnetic permeability is similar, the magnetic circuit outside is surrounded by aluminum alloy, and the magnetic permeability of the aluminum alloy material is much lower than the magnetic circuit structure permeability; therefore, a parallel boundary is adopted to achieve magnetic isolation.

To determine the average magnetic induction intensity in the damping channel, 20 points were evenly spaced in the controllable channel of the magnetorheological fluid, as shown in Figure 7. The average magnetic induction intensity of 20 points can be approximated to obtain the average magnetic induction intensity value of the damping channel.

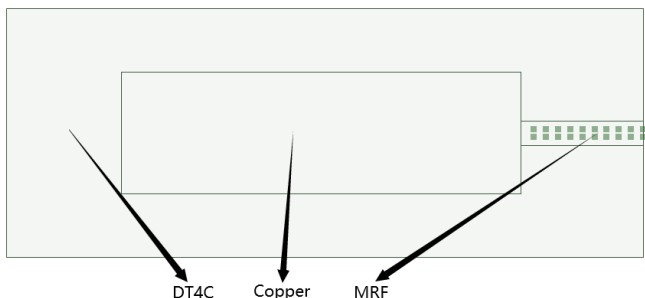

**Figure 7.** Damping channel node diagram.

When the excitation current is the operating current $I$ = 1.5 A, the magnetic induction intensity distribution at the magnetic circuit and damping channel is as shown in Figure 8. The maximum magnetic induction intensity of the entire magnetic circuit reached saturation at the corner of the core coil slot, and the average magnetic induction intensity was concentrated between 1.3~1.8 T. The magnetic induction intensity at the channel corresponding to the effective magnetic pole varied relatively uniformly with the radial distance, and the average magnetic induction intensity was approximately 0.43 T. Evidently, in the original structure, the magnetic field strength of DT-4C is saturated, while the magnetic induction strength in the magnetic circuit is far from saturated, and the utilization rate of the magnetic circuit on the magnetic field can be further optimized.

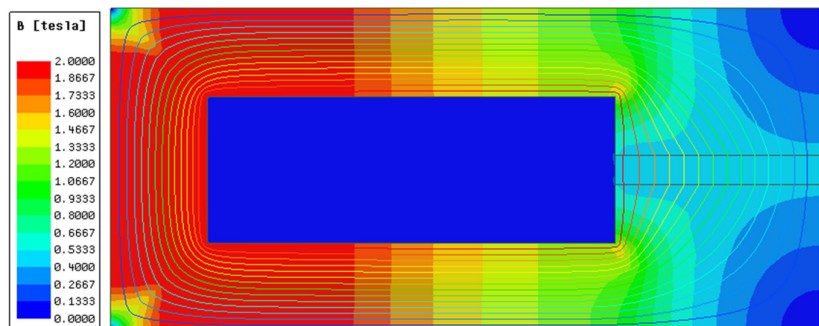

**Figure 8.** Cloud chart at the current of 1.5 A.

Bringing the magnetic induction data from Figure 8 into Equation (8), a zero-field damping force and maximum damping force of 55.42 and 248.42 N can be obtained, respectively.

### 2.4. Model Verification and Correction

The MRF mount was processed and assembled according to the size of the magnetic circuit, and the test samples and equipment are shown in Figures 9 and 10. The weight of the mount was 2789.8 g, and the test equipment consisted of the excitation power supply and a vibration test bench of an enterprise.

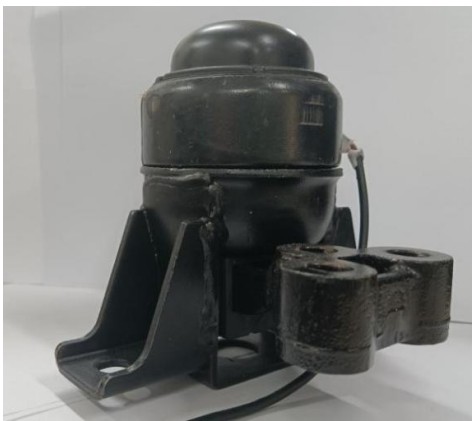

**Figure 9.** MRF mount.

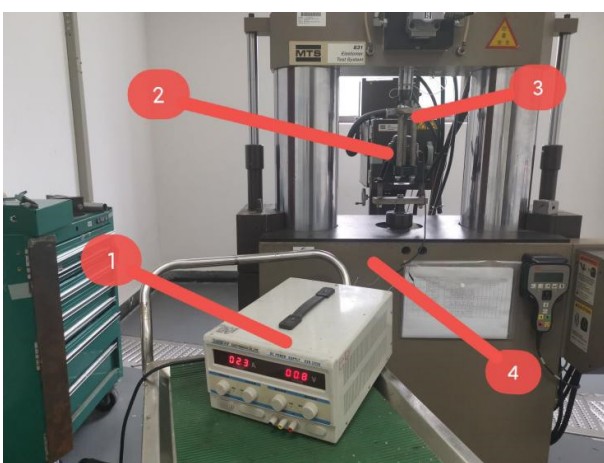

**Figure 10.** Test instruments. (**1**) Power supply. (**2**) MRF mount. (**3**) Fixture. (**4**) Performance test rig of MRF mount.

The test data were collected when the stroke was 1 mm, preload was 1000 N, frequency range was 1–50 Hz, interval was 1 Hz, and current was 0~1.8 A. These are shown in Figures 11 and 12, respectively. The dynamic stiffness and damping characteristics of the magnetorheological mounts exhibit obvious nonlinear relationships with excitation frequency. Figure 11 shows that, in the 115 Hz frequency range, the suspension stiffness increased sharply with the increasing frequency and tended to level off after 15 Hz. Under the condition that the excitation frequency and amplitude are constant, the dynamic stiffness of the mount increases with an increase in current. Figure 12 shows that, at the same excitation frequency, the magnetorheological fluid suspension damping increases with an increase in the excitation current instead of linearly; under the same excitation current, the damping decreases nonlinearly with an increase in the excitation frequency, and the effect of the current on the damping decreases with an increase in frequency.

The mount was subjected to forced vibration in the vertical direction, excited by a single-frequency sinusoidal displacement. According to the theory of forced vibration [34] suspended under the displacement excitation of forced vibration, the restoring force comprises elastic and damping forces.

$$X = X_0 \sin \omega t \tag{12}$$

$$F = F_0 \sin(\omega t + \Delta \varphi) \tag{13}$$

where $X_0$ is the amplitude of the displacement, $\omega$ is the frequency of the excitation angle, $F_0$ is the amplitude of the restoring force, and $\Delta \varphi$ is the phase difference.

$$F = KX + C\dot{X} \tag{14}$$

$$K = K^* \cos \Delta\phi = F_0 \cos \Delta\phi / X_0 \tag{15}$$

$$C = K^* \sin \Delta\phi / \omega = F_0 \sin \Delta\phi / (\omega X_0) \tag{16}$$

where $K$ is the elastic stiffness, $K$ is the dynamic stiffness, $C$ is the damping, and $F$ is the restoring force.

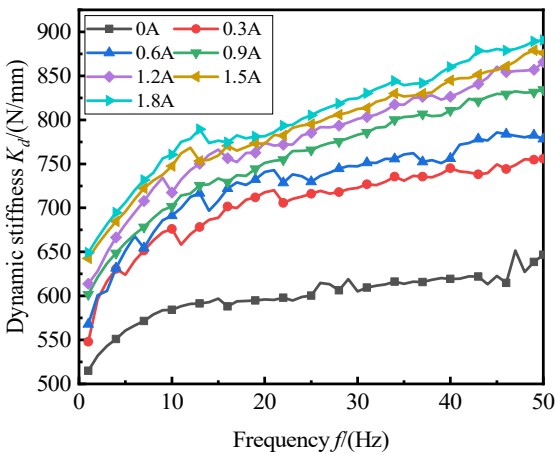

**Figure 11.** Dynamic stiffness of the MRF mount.

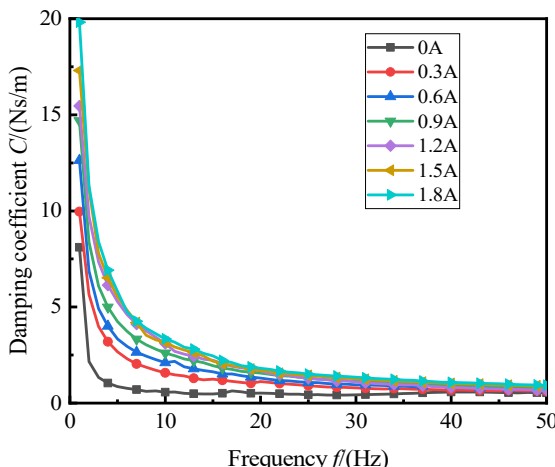

**Figure 12.** Damping coefficient of the MRF mount.

This is shown in Figures 13 and 14, which show the time domain results of the recovery force and control force of the mounting force under different currents when the preload was 1000 N, excitation frequency was 10 Hz, and amplitude was 1 mm.

The results of the simulation and experiment were compared, as shown in Figure 15. As can be seen from the figure, the growth trends of the testing and simulation were basically the same. The recovery force deviates more from the simulated values because the interaction between the fluid and solid was not considered in the simulation estimation in this study. The change in the controllable force is mainly due to the influence of the magnetic circuit, and the other structures have little effect on it.

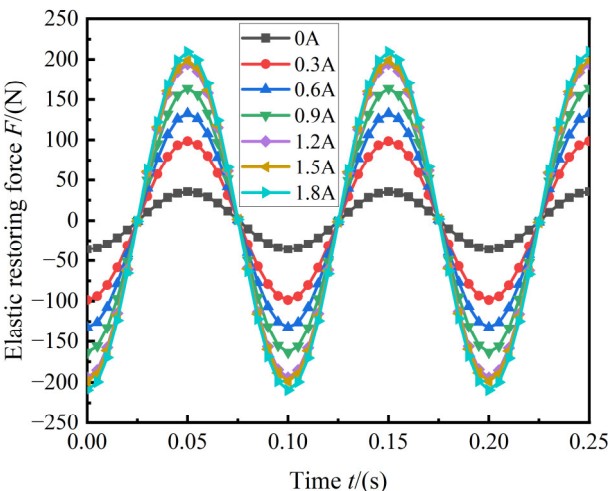

**Figure 13.** Elastic restoring force of the MRF mount.

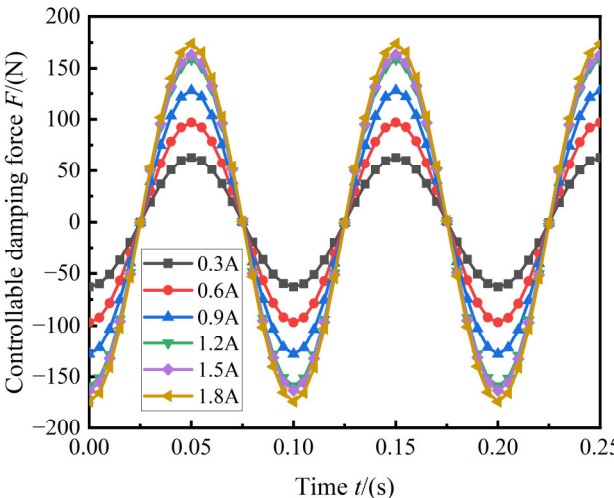

**Figure 14.** Controllable force of the MRF mount.

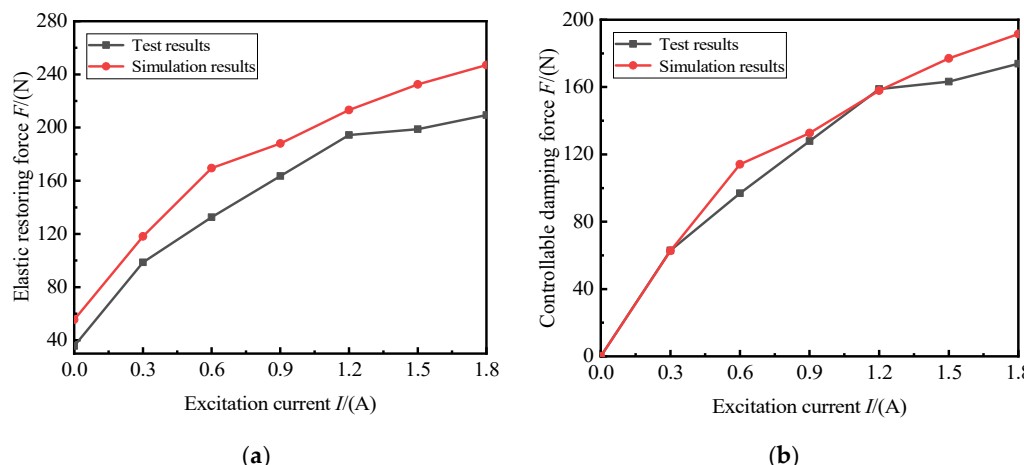

**Figure 15.** Comparison of simulation and experimental results: (**a**) comparison of elastic restoring force; (**b**) comparison of coulomb damping force.

We considered the predictive model used in this study, which is a steady-state model. Therefore, to improve the accuracy of the study, it is necessary to study the error at different

frequencies. Figure 16 shows the controllable force estimated by the test and model at different frequencies at 1.5 A. The fluid model used to predict the damping force in the simulation is for illustrating the damping force under the ideal condition of the fluid. With an increase in the frequency, the flow rate of the fluid flowing through increases, and the damping force increases. However, in practice, the fluid is far from that of the flow rate in theory, and even dynamic hardening occurs at high frequencies.

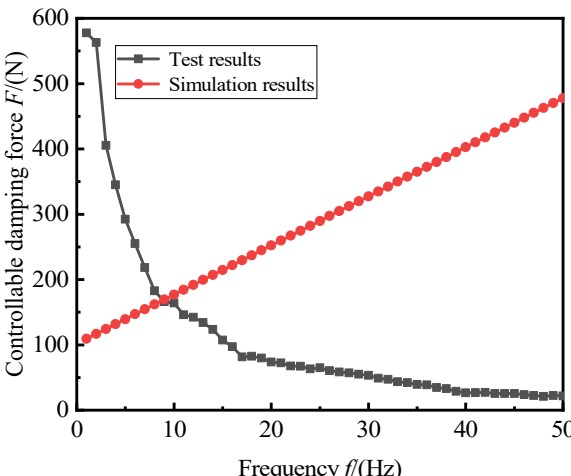

**Figure 16.** Comparison of controllable force at different frequencies.

In the follow-up study, the design mount at 1.5 A basically reached magnetic saturation, and all the simulations and optimizations were carried out under this premise. Therefore, to further improve the accuracy of the model, the function suitable for the 1.5 A current of this model is fitted.

$$\Delta F = F_c - F_0 + \varphi_f \tag{17}$$

$$\varphi_f = 524.36 - 87.1929x + 4.1879x^2 - 0.0952x^3 + 0.0008x^4 \tag{18}$$

where $\Delta F$ is the corrected controllable force, $F_0$ is the damping force of the zero magnetic field, and $\varphi_f$ is the correction function of the model at 1.5 A.

## 3. Analysis of Sensitivity

To explore the sensitivity of each structural parameter to the damping force, Isight software and Maxwell were used to jointly simulate the sensitivity analysis under 10 Hz of the common working conditions of the magnetorheological mount. Taking the size of the suspended magnetic circuit structure as the design variable, the number of design variables is 7, and the viscous damping force and coulomb damping force of the mount are selected as the responses. Owing to the advantages of the effective spatial filling ability of the Latin hypercube and the fitting nonlinear response, the Latin hypercube sampling method was selected in this study.

The selection of the value range includes the maximum number of possible size parameters that fits the space size limit of this mount. Table 2 shows the selected optimization design variables and their respective value ranges.

To determine the most appropriate number of the sampling points, different sampling points were selected for the analysis and their influence on the simulation was studied. The simulation results are shown in Figure 17, where the red curve represents a negative effect. With the change of the number of sampling points, the influence of each structural parameter on the damping force fluctuates within a certain range, the influence of each structural parameter on the viscous damping force from the number of sampling points starts at 1400, and the fluctuation amplitude of each structural parameter on the viscous

damping force is less than 0.5%; therefore, it is considered that it tends to converge, and the number of sampling points is selected to be 1400.

**Table 2.** Magnetic circuit structure parameter value range.

| Variables | Lower Bound/(mm) | Upper Bound/(mm) |
|---|---|---|
| $R_1$ | 5 | 12 |
| $R_2$ | 1 | 5 |
| $R_3$ | 10 | 18 |
| $R_4$ | 5 | 12 |
| $L_1$ | 1 | 5 |
| $L_2$ | 1 | 3 |
| $H$ | 1 | 5 |

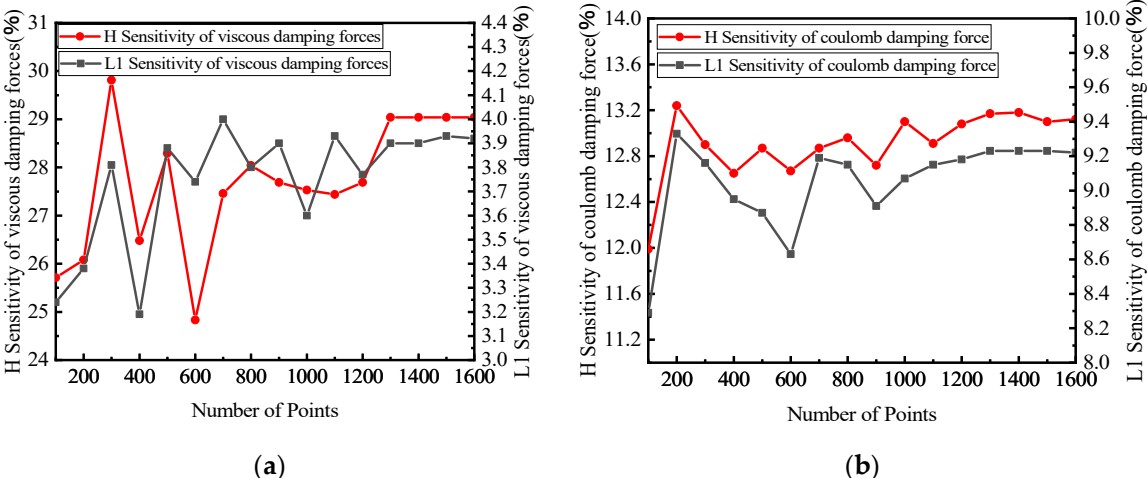

(**a**)　　　　　　　　　　(**b**)

**Figure 17.** Sensitivity convergence of some parameters: (**a**) sensitivity of some parameters to viscous damping forces; (**b**) sensitivity of some parameters to coulomb damping force.

Figure 18 shows the sensitivity of each structural parameter to damping force. Because of the excessive data of each structure, if all the data are plotted in the figure, it will affect the display effect. Figure 18a below only shows the data with a sensitivity of more than 2.5%, and Figure 18b shows that the data are denser and only shows a sensitivity of more than 3.5% (blue represents positive effects, and red represents negative effects).

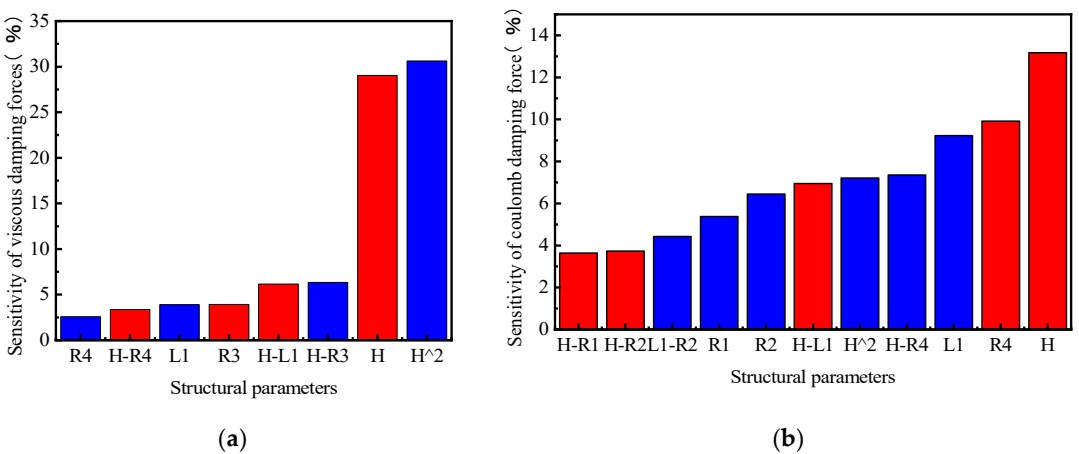

(**a**)　　　　　　　　　　(**b**)

**Figure 18.** Sensitivity analysis of partial results: (**a**) sensitivity of design variables to viscous damping forces; (**b**) the sensitivity of the design variables to coulomb damping force.

Combined with the above sensitivity analysis results, among the seven structural parameters, $L_2$ had the least sensitive effect on the damping force. The dimensions that determine the cross-sectional area of the inertial channel, namely, $H$ and $R_4$, were the most sensitive. Combined with the theoretical derivation formula, $R_4$ and $H$ are directly related to the cross-sectional area of the inertial channel. $R_2$ and $L_1$ are directly related to the coil, and the sizes of the four parameters $R_1$, $R_2$, $R_3$, and $R_4$ determine the length of the inertial channel. The output damping force of the MRF mount is related to the structural parameters of the magnetic circuit and the magnetic induction strength of the MRF in the effective damping channel. The viscous damping force of the mount is determined only by the parameters of the magnetic circuit structure, whereas the coulomb damping force is determined by the structural parameters of the magnetic circuit and the magnetic induction strength of the MRF in the effective damping channel. Both the simulation results and theories show that $L_2$ has little effect on the viscous and coulomb damping forces. In the results that are not shown here, the sensitivity of the viscous damping force with L2 was up to 0.84%, and the sensitivity of the coulomb damping force with L2 was up to 1.09%. The effect of $L_2$ on the damping force was negligible, that is, less than 4%.

## 4. Multi-Objective Optimization Design of Magnetic Circuits

The optimization design, which is presented using Maxwell and Isight for the co-simulation, is shown in Figure 19. The determination of the three optimization elements of the optimization objectives, design variables, and constraints is extremely important for optimization. Additionally, the main parameters of the algorithm are changed to achieve convergence. The Pareto frontier is analyzed, and the optimal individuals that best meet the requirements are selected through fuzzy set theory and experience.

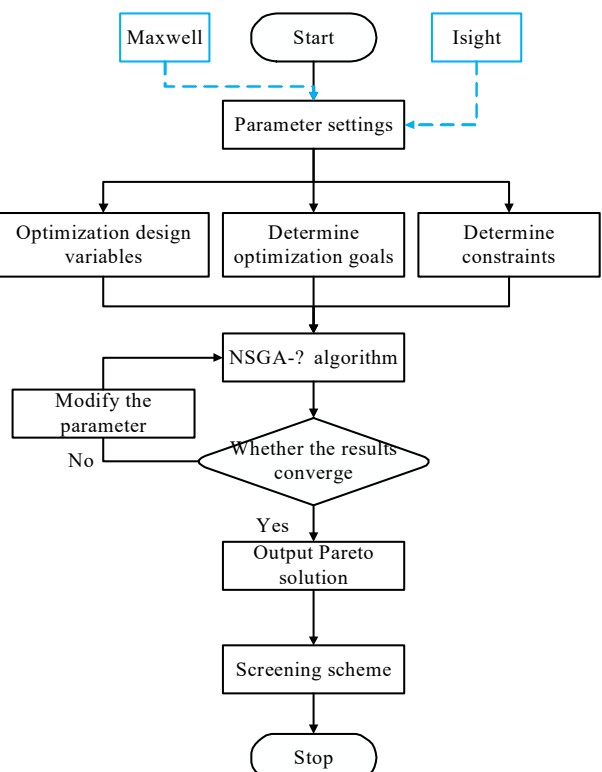

**Figure 19.** Working flow of optimal design.

The vibration isolation performance of the mount is determined by the viscous damping force and coulomb damping force. The greater the output damping force, the stronger is the absorption effect on the vibration. The adjustable damping coefficient reflects the ability of the MRF mount to adjust the damping force range; furthermore, the larger the

adjustable damping coefficient, the wider the working conditions are for which the damper can be applied. Therefore, the maximum output damping force and damping adjustable coefficient are the optimization goals.

The damping adjustability factor is defined to evaluate the adjustability of the mount:

$$\beta = \frac{\Delta F}{F_0} \tag{19}$$

The objective function can be expressed as:

$$\max F(x) = [F_c, \beta]^T \tag{20}$$

The structural parameters of the sensitivity analysis were selected as design variables after comprehensive consideration of the magnetic circuit design requirements, working characteristics, and mechanical characteristics of the MRF mount. Simultaneously, combined with the sensitivity analysis results, the design variables are set as $H$, $L_1$, $R_1$, $R_2$, $R_3$ and $R_4$, which are six structural parameters, as listed in Table 3.

**Table 3.** Multi-objective optimization range of structural parameters.

| Optimize Variables | Lower Bound/(mm) | Upper Bound/(mm) |
|:---:|:---:|:---:|
| $R_1$ | 5 | 12 |
| $R_2$ | 1 | 5 |
| $R_3$ | 10 | 18 |
| $R_4$ | 5 | 12 |
| $L_1$ | 1 | 5 |
| $L_2$ | 2 | 2 |
| $H$ | 1 | 5 |

Combined with the dimensional design requirements of this mount and the actual material characteristics, the magnetic induction strength of the design should be less than the saturated magnetic induction strength of the MRF and DT-4C. The constraint formula is as follows:

$$\begin{cases} R_1 + R_2 + R_3 + R_4 = 32.3 \\ 2 \times (L_1 + L_2) + H = 14 \\ R_3 \times (2 \times L_2 + H) > 80 \\ \max(B_1, B_2, \cdots, B_{20}) < 0.54 \\ B_{DT4C} < 1.66 \end{cases} \tag{21}$$

The NSGAII algorithm has the advantage of a good exploration performance. To obtain the global optimal value, the NSGA II algorithm was used to perform global search optimization on the entire design space of the optimized variables [35]. The crossover rate, crossover distribution index, and mutation distribution index were set to 0.9, 10, and 20, respectively [36]. As shown in Figure 20, with an increase in the population size and maximum evolutionary algebra, the optimized maximum viscous damping force and coulomb damping force gradually begin to converge, and the subsequent fluctuation range is less than 5 N, which is considered to have converged. Therefore, the population size and maximum evolutionary algebra were selected to be 320 and 20, respectively.

After calculation, 101 Pareto optimal solutions for the suspended magnetic circuit were obtained; the Pareto front is shown in Figure 21. Obviously, the inverse relationship between the two objective functions of the output damping force and damping tunable coefficient is not guaranteed to simultaneously meet the maximum design goal. Therefore, it is particularly important to select the individual in the Pareto noninferior solution as the optimal individual. Table 4 lists the partial solutions for Pareto's optimal solution set domain.

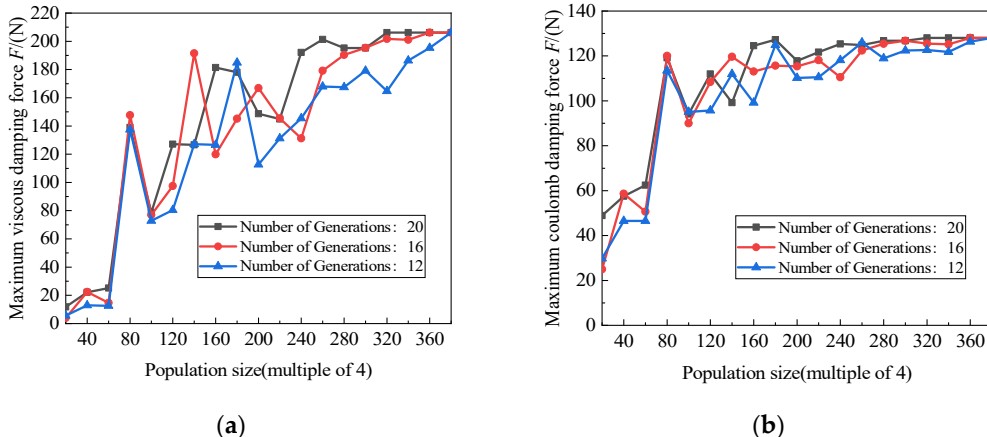

(**a**)　　　　　　　　　　　　(**b**)

**Figure 20.** Optimized convergence of different parameters: (**a**) optimized maximum viscosity damping force; (**b**) optimized maximum coulomb damping force.

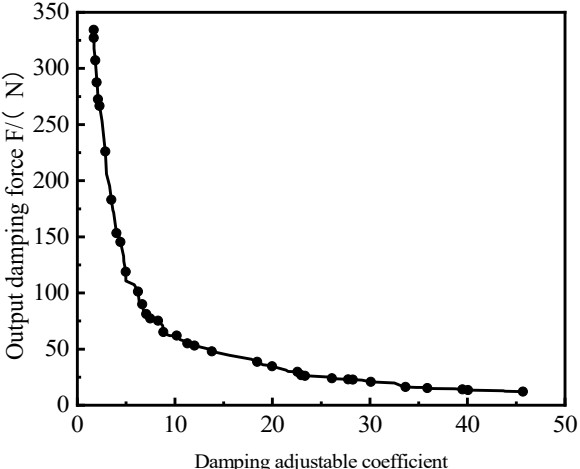

**Figure 21.** Pareto optimal solution set.

**Table 4.** Partial Pareto solution set.

| Number | $H$ | $L_1$ | $R_1$ | $R_2$ | $R_3$ | $R_4$ |
|---|---|---|---|---|---|---|
| 1 | 1.01 | 3.22 | 5.82 | 3.74 | 17.46 | 5.01 |
| 2 | 1.03 | 2.26 | 6.80 | 3.44 | 16.62 | 5.22 |
| 3 | 1.01 | 2.82 | 5.05 | 4.68 | 16.38 | 5.08 |
| 4 | 1.01 | 2.52 | 5.26 | 4.59 | 16.33 | 5.03 |
| 5 | 1.26 | 3.40 | 6.07 | 3.07 | 16.06 | 6.96 |
| 6 | 1.17 | 3.40 | 6.07 | 2.04 | 16.23 | 7.84 |
| 7 | 1.04 | 2.96 | 6.09 | 3.02 | 15.95 | 5.93 |
| 8 | 1.04 | 2.86 | 6.14 | 2.93 | 16.24 | 5.88 |
| 9 | 1.05 | 3.38 | 6.07 | 2.04 | 16.23 | 7.87 |
| 10 | 1.04 | 3.43 | 6.07 | 3.06 | 16.06 | 7.06 |
| 11 | 1.01 | 2.53 | 7.93 | 1.33 | 16.08 | 5.93 |
| 12 | 1.04 | 2.96 | 6.16 | 3.02 | 16.88 | 5.52 |
| 13 | 1.04 | 2.31 | 6.80 | 3.44 | 16.61 | 5.26 |
| 14 | 1.03 | 2.26 | 6.80 | 3.44 | 16.62 | 5.22 |
| 15 | 1.01 | 2.64 | 6.00 | 4.69 | 16.38 | 5.03 |

To better solve the multi-objective optimization problem, fuzzy set theory was used to perform further selection among the solution sets. The membership function $\mu_i$ and domination function $\phi_k$ are defined as follows:

$$\mu_i = \begin{cases} 0, & f_i \leq f_{\min i} \\ \frac{f_i - f_{\min i}}{f_{\max i} - f_{\min i}}, & f_{\min i} < f_i < f_{\max i} \\ 0, & f_i \geq f_{\max i} \end{cases} \tag{22}$$

$$\varphi_k = \sum_{i=1}^{M_O} \mu_i^k \Big/ \left( \sum_{j=1}^{M_P} \sum_{i=1}^{M_O} \mu_i^j \right) \tag{23}$$

where $f_{\max i}$ and $f_{\min i}$ are the maximum and minimum values of the $i$ optimization target, respectively; $f_i$ is the value of the $i$ optimization goal; $M_p$ is the number of solutions contained in the Pareto set; and $M_o$ is the number of optimization targets.

We calculated the domination function of each individual in the Pareto optimal solution set; the larger the $\varphi_k$, the better is the overall performance of the solution. As shown in Figure 22, the dominance value of individual No. 82, with the largest dominant function value, was approximately 0.0170. Individuals 7, 91, and 92 were similar to their dominant values, and individual 7 was excluded due to small damping forces. Individuals 91 and 92 were similar in size, which is limited by the manufacturing process in the actual manufacturing and can be regarded as a result. Compared to individual No. 82, the average magnetic induction intensity of individual No. 82 at 1.5 A current is approximately 0.512 T, whereas No. 91 and No. 92 are only 0.487 T, which is obviously higher than the magnetic induction intensity of individual No. 82, which is regarded as the optimal individual that best meets the requirements.

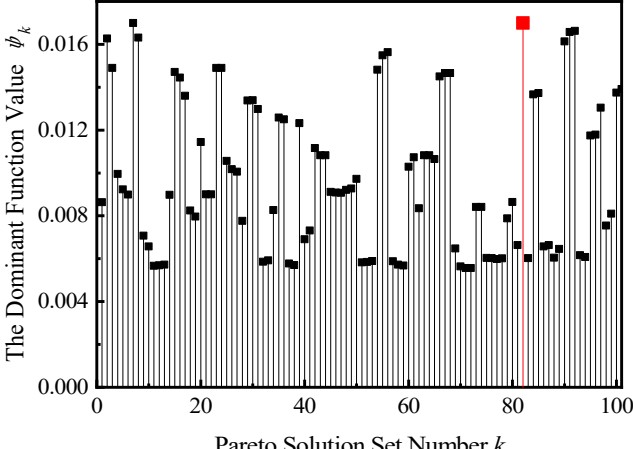

**Figure 22.** Value of the dominant function of the Pareto optimal solution set.

## 5. Comparison of Results

Table 5 shows the magnetic circuit structure before and after the multi-objective optimization. Owing to the limitations of the actual manufacturing process, the optimization results are only taken to one decimal place. Table 6 compares the results when the excitation current was 1.5 A before and after optimization. After optimization, the theoretical output damping force of the MRF mount was 336.19 N, which was 44.64% higher than that before optimization. The controllable force and zero magnetic field damping force were also increased by 43.70% and 47.64%, respectively, and the adjustable damping coefficient was reduced by −2.50%. The description shows that the optimization method has an obvious effect on improving the damping performance of the mount and ensures that the MRF mount has a wider adjustable ability. The adjustable damping coefficient is reduced,

which also reflects the inverse trend of the two optimization goals from the side, which is consistent with the above analysis of the optimization solution set.

**Table 5.** Optimize the front and rear structure parameters.

|  | $R_1$ | $R_2$ | $R_3$ | $R_4$ | $L_1$ | $H$ |
|---|---|---|---|---|---|---|
| Before optimization | 7 | 3.3 | 14 | 7 | 3 | 1 |
| After optimization | 6.0 | 4.7 | 16.4 | 5.0 | 2.6 | 1.0 |

**Table 6.** Optimization results comparison.

|  | Output Damping Force (N) | Controllable Force (N) | Zero Magnetic Field Damping Force (N) | Adjustable Coefficients |
|---|---|---|---|---|
| Before optimization | 232.44 | 177.02 | 55.42 | 3.19 |
| After optimization | 336.19 | 254.37 | 81.82 | 3.11 |
| Rate of change (%) | 44.64 | 43.70 | 47.64 | −2.50 |

As shown in Figure 23, when the excitation current is the maximum working current, that is, I = 1.5 A, the magnetic induction intensity at the channel corresponding to the effective magnetic pole varies relatively evenly with the radial distance, and the average magnetic induction intensity is approximately 0.51 T. The MRF was close to saturation and significantly higher than that before optimization.

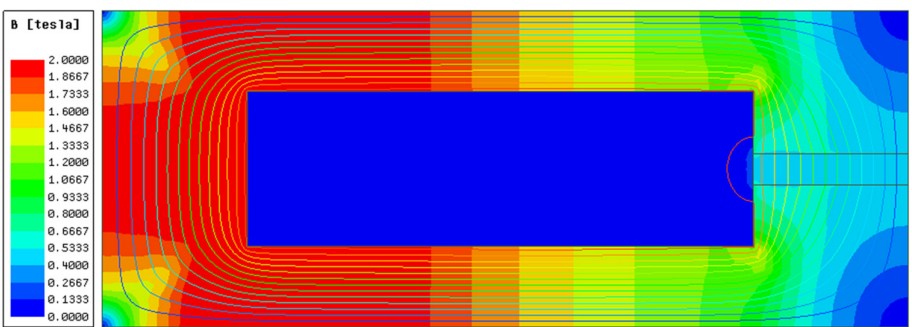

**Figure 23.** Cloud chart of magnetic induction intensity and magnetic lines.

Figure 24 shows a comparison between before and after the optimization. After optimization, the magnetic induction intensity increased significantly under the same current. The average magnetic induction intensity at the optimized damping channel is approximately 0.43 T at 1.5 A, the average magnetic induction intensity at the optimized damping channel is approximately 0.51 T, and the average magnetic induction intensity is increased by 18.60%. The enhancement of the magnetic induction strength indicates that the utilization rate of the magnetic circuit is improved, effectively reducing energy consumption. The controllable force increases with an increase in current, and the controllable force is greater after optimization. The controllable force increases from 1.5 A to 1.8 A before optimization by 8.20%, and only by 2.26% after optimization. This indicates that the damping performance is maximized when the optimized mount is 1.5 A, which meets the design requirements.

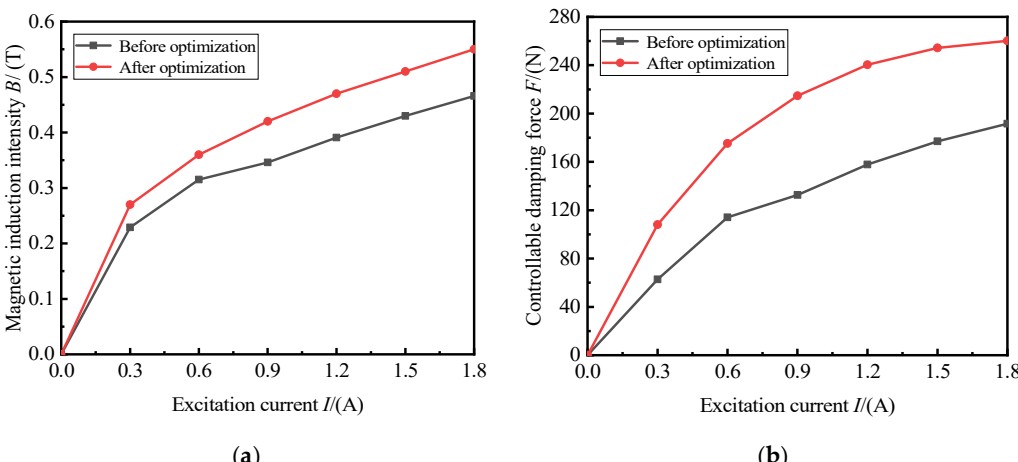

**Figure 24.** Simulation comparison before and after optimization: (**a**) comparison of magnetic induction intensities; (**b**) comparison of the controllable force.

## 6. Conclusions

In this study, a radial-channel-type MRF mount was designed, simulated, fabricated, and tested. Based on the proposed analysis method, the magnetic circuit and magnetic field distribution of the MRF mount were theoretically analyzed, and the damping performance was predicted by the mathematical model of the proposed MRF mount. Combined with a multi-objective genetic algorithm, the structure of the magnetic circuit was optimized. The results show that:

1.  Simplifying the magnetic circuit structure into a two-dimensional axisymmetric model can reduce the calculation time of the model and effectively improve the calculation speed during joint simulation optimization by ensuring accuracy. At the same time, most fluid models used today are steady-state models with certain errors, and the accuracy of the model can be further improved by fitting the extended equation. The experimental simulation comparison method described in this paper can compare the errors of theoretical and practical models from different frequency perspectives.
2.  Combined with the theoretical analysis and simulation verification, the core depth $L_2$ has the least influence on the damping force. The results show that the influence of $L_2$ on the coulomb damping force and viscous damping force is much less than 4%, which can be ignored in the structural optimization. The coulomb damping force is most correlated with the cross-sectional area and length of the inertial channel, that is, the height of the inertial channel $H$ and the width of the inertial channel $R_4$.
3.  The test results indicate that the simulation model is consistent with the change trend in the actual mount damping force, and the controllable force values are similar. After optimization, the average magnetic induction strength is increased by 18.60%, and the output damping force reaches 336.19 N. When the current is increased from 1.5 A to 1.8 A, the controllable force increased by only 2.26%, indicating that the damping performance was fully exerted at 1.5 A, which is in line with the design expectations.

**Author Contributions:** Conceptualization, J.F. and C.H.; methodology, J.F. and R.S.; software, C.H.; validation, X.-Q.L. and M.C.; formal analysis, Z.C.; investigation, B.C.; resources, X.-Q.L.; data curation, M.C.; writing—original draft preparation, C.H.; writing—review and editing, R.S.; visualization, Z.C.; supervision, B.C.; project administration, C.H.; funding acquisition, J.F. All authors have read and agreed to the published version of the manuscript.

**Funding:** This study was sponsored by the Natural Science Foundation of Chongqing, China (cstc2020jcyj-msxmX0226).

**Institutional Review Board Statement:** Not applicable.

**Informed Consent Statement:** Not applicable.

**Data Availability Statement:** Not applicable.

**Acknowledgments:** The authors gratefully acknowledge the financial support for this work from the Natural Science Foundation of Chongqing, China (cstc2020jcyj-msxmX0226). The authors thank the editors and reviewers for their valuable comments and constructive suggestions.

**Conflicts of Interest:** The authors declare no conflict of interest.

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
