# Peer review of "Multi-Objective Optimization of Magnetorheological Mount Considering Optimal Damping Force and Maximum Adjustable Coefficient"

_machines, doi:10.3390/machines11010060_

Round 1
Reviewer 1 Report
The authors present a multiobjective optimization of a MR-based system. The methods used have to be better introduced and discussed, particularly regarding the design needs (e.g. max and min Damping Force). Information about instruments and software used must be added.
I think the paper must be improved to consider the following points:
-Line 140: “The deformation of the liquid”, please check
-Line 159-164: “To determine the average magnetic induction intensity in the damping channel, take (?) 20 points at a uniform interval at the controllable channel of the MRF … approximated.” Unclear sentence, please simplify the structure of the phrase.
-Figure 6: Please indicate the part of the structure, e.g. copper, etc.
Line 171, Please check the value: 1.31.8T
-Figure 12 and Figure 13. Confirm that these data are obtained experimentally. In this case, I think the authors should comment on the presence/absence of scattering in the experimental data.
-Figure 15: I think this figure needs a more explicit comment by the authors.
-Line 233: “variable”? Please check
-Please add information about the software used for numerical simulation and optimization.
-Line 237 – Please insert a Figure of the system that introduces the design variables selected and describe them in the text. Please also add a description of the reasons for choosing the ranges as they are decisive for the identification of the optimisation solution.
-Line 255 and line 266:“Combined with the above sensitivity analysis results, among the 7 structural parameters, L2 has the least sensitive effect on the damping force.” “Both the simulation results and theories show that L2 has little effect on viscous damping force and controllable damping force. As shown in Figure 17, the effect of L2 on the damping force is less than 4% and can be ignored.”
I can not find L2 in Figure 17! Please check
-Table3: caption has to be corrected and what is the difference between Table 2 and 3? They are currently identical.
-Line 292: NSGAII, please add a literature reference for this method
-line 295 “default values of 0.9, 10 and 20, respectively.” Why these are default values? Who introduces it in literature?
-Line 303: Do the authors means Pareto frontier? Please specify
-Line 321: It is known that little difference in dominant function value results is usually negligible in their implications for design choice. So other factors (such as: costs, lightweight, MR fluid quantity, manufacturing complexity, etc.) guide the decision within the group of high-potential solutions. The authors should better present the reasons why they chose to identify the number 82 and discard other points with a slightly lower value.
Page 16: The design of the system should, in my opinion, consider the damping force in the absence of a magnetic field as invariant since, from this value upwards, we can act with current control. Values below this are, in fact, unattainable in use. Why do the authors not constrain the optimisation of the system so that the value of this damping force is approximately constant and set to the needs of the problem? Otherwise, the solution obtained is optimised for the maximum damping force but out of specification for the minimum one.
-Conclusions have to be aligned with results explicitly described in the paper. I note that point 2 is based on the evaluation of L2 impact that is currently not present clearly in the results.
-In my opinion, Point 1 of the conclusion presents obvious results about asymmetric architecture for the system that does not depend on the information presented in the paper) as a result of the paper.
Reviewer 2 Report
The manuscript presents multi-objective design optimization of an MR mount considering the optimal damping force and maximum adjustable coefficient. Based on the results, the magnet core depth under the radial channel MRF mount structure is the least influential factor on damping force. A change in the inertial channel height will improve damping performance.
General comment: There are many typos and grammatical errors throughout the manuscript. It is highly recommended to re-read and re-revise the manuscript in order to make it suitable for the readers. It cannot be followed with the current version. The abstract also needs a thorough revision.
Authors delivered an incremental work on design optimization of MR mounts. The outcomes of the current study in terms of stiffness and damping variability as well as damping forces were not compared with the state-of-the art works on MR mounts. Therefore, it is difficult to find the major contribution of this work in terms of materials, modeling, and characterization. Based on this consideration, I cannot strongly support this manuscript unless there is novel contribution for the field.
There are, also, several major concerns related to the technical perspectives that need to be addressed, such as:
1. The sentences and paragraphs in the introduction are not properly connected.
2. The contribution of the present work has not been clearly reflected in the introduction.
3. The focus of the paper is on the MR mount; however, there is no discussion on the MR mount in the introduction!
4. It is mandatory to elaborate on the most state-of-the art MR mount in the introduction.
5. In the section 2.1, why the magnetic core assembly is selected as an axisymmetric structure?
6. What is DT-4C? Should be clearly mentioned
7. The nonlinear B-H curve of MR fluid should be added.
8. The equation 1 is not physically correct. The magnetic field induction should be modeled as an exponential form to show the saturation phenomenon.
9. In Eq. 1, when B is zero, what is the physical meaning of the 1.09 for the shear yield stress? The yield stress at zero field should be zero or not? Why?
10. The parameters in the presented equations are not defined and not explained.
11. Why was the number of coils obtained as 240? What is the diameter of the wire? AWG?
12. Is there any leakage flux and/or fringing flux in the magnetic circuit design?
13. Did authors consider the total weight of the MR mount? Any physical constraint?
14. How long the MR mount’s magnetic circuit can operate at 1.5A?
15. The total weight of the MR mount should be added to the paper.
16. What kind of software is used in section 2.3?
17. Why the preload has been considered as 1kN? How much is the corresponding displacement under 1kN preload?
18. How were the damping and stiffness derived?
19. The force-displacement curves should be added to the paper.

Reviewer 3 Report
General:
The manuscript aims to present a multi-objective optimisation study of an MR mount. The dimensions of the magnetic circuit in the MR mount are optimised based on FE modelling of magnetic field.
Overall, in terms of the clearness of contents, the methods applied need to be extensively explained. It is not clear which software has been used or the governing equations and many other details, as shown in the detailed comments.
In terms of novelty, optimisation studies regarding the dimensions of magnetic circuit of MR devices are extensively found in the literature, as shown in the Ref [15-20] cited by the authors. Most of these references present optimisation models that seem to more advanced compared to the model presented in the current manuscript. The authors need to explain how their model is different.
The manuscript suffers very weak academic writing. The literature presented in the introduction are not critisised. The methods and the characteristics of the damper are not well-described, and most results are not justified. My detailed comments are as follows:
Introduction:
1. “Magnetorheological Fulid is a new type of intelligent fluid”. MR fluids were invented in 1947. Also, there is a spelling mistake.
2. More details need to describe the work in Ref. [15-20]. These studies presented optimisation techniques. So, they merit more critising in order that the reader can capture the gap which the current manuscript will fulfill.
3. Line 75:77: If that is the conclusion of LR, so the model needs to be coupled with CFD. I don’t think that the computational time is a big issue to hinder the implementation of this type of models.
Section 2:
4. The construction of the mount is not well-presented. The complete paths of MR fluid and the chambers in both strokes should be seen. The authors can use zoom on some locations and dashed lines to indicate the relative positions of some parts.
5. The “connecting rod” is stated and not seen on the fiqures.
6. Line 106: Is there a name of the MRF used? Is it commercially available?
7. What information does Figure 4 provides? I believe that taw_y can be plotted vs B rather than H in Fig. 3, and this will link with the curve fiiting equation.
8. Line 129: What is the meaning of “ liquids are symmetrical?”
9. Line 148: How the working area was calculated?
10. Line 162: There is no need to determine the average magnetic field using 20 Pts. It can be calculated over this area of the mesh using the FE solver / post processing.
11. Section 2.3 : There is no explanation of what software has been used, governing equations, magnetic insulation, no of coil turns, mesh data, mesh dependence solution, etc .
12. Line 171: Typo in magnetic field value.
13. What are the details of the magnetic material used? It seems that DT-4C is a commercial name. It does not provide information about the alloy and its constituents.
14. There is no comment on Figure 8, 9, 10 and 11. Same in most figures in the manuscript. More information should be provided.
15. Line 199-201: Providing equations may be necessary here to interpret Figures 12 and 13.
16. It is not clear from figure 14 how the simulation results were obtained.
17. Simulation and experimental results in Fig. 15 tell different stories. There is no comment provided.
Section 3:
18. There is no regime of the change of parameters in Fig. 16. This should be justified.
19. What is the tools used for optimisation? What are the equations?
20. Optimisation studies are frequently accompanied by surface plots. It will be useful to see some of these plots.
Section 4:
21. More explanation should be provided regarding the Multi-objective Optimization and Pareto's optimal solution.
Section 5:
22. What is the meaning of cloud chart?
Round 2
Reviewer 1 Report
The authors have correctly improved the paper, which is now suitable for publication.
Author Response
Thank you very much for your recognition and trust.Your comments give us a lot of new ideas to make our articles better. To make the article more believable, we have added a little detail again. Finally, thank you again for your comment, it is very informative for my study and research, and I will keep your thoughts in mind in my future work.
-Line 173-176:Added description and formula for coil turns.
-Line 177:The source of the current density value range.
-Line 184-186:Added descriptions of the governing equations, meshes, etc.Make manuscript content easily verifiable to confirm its authenticity.
-Line 449-451:Refine the description of the conclusion.
-Line 454 and 455:Unified article description.

Reviewer 2 Report
Please check that all the comments have been addressed in the manuscript properly.
Some of the comments responses still need to be added to the manuscript.
For instance:
1- The answer to the comment 11.
2- The authors replied that “the allowable range is J  5~12A/mm2.”. It is recommended to add a reference for this claim.

Author Response
Thank you very much for your comments, which gave us a lot of new thinking and made our articles better. We checked again and all your comments made sure they were all replied. At the same time, we agree with your comments on question 11. A more detailed description of the number of coil turns is necessary. We have revised parts of the manuscript again.
-Line 173-176:Added description and formula for coil turns.
-Line 177:The source of the current density value range.
-Line 184-186:Added descriptions of the governing equations, meshes, etc.Make manuscript content easily verifiable to confirm its authenticity.
-Line 449-451:Refine the description of the conclusion.
-Line 454 and 455:Unified article description.

Reviewer 3 Report
The manuscript still suffers serious flaws in terms of academic writing, structure of sentences, presentation of ideas, interpretation of the methods. The novelty of the work is not clear. The FE simulation of magnetic field is frequently found in the literature. The analytical model is also reported in different studies. Besides, it does not describe the performance of the mount at high frequencies, as declared by the authors in Line 88-96. The method used in optimisation model is also reported in many studies. Most importantly, it is based on an analytical model whose accuracy is doubtable in terms of the evaluation of output force.
Author Response
Thank you very much for your comments, which gave us a lot of new thinking and made our articles better.Regarding analytical models, most of today's research also uses pseudohomeostatic theoretical models, which are similar to ours. I have to admit that there are differences and errors between such a model and the actual one, and our model is no exception. However, we believe that the method mentioned in this article to compare simulation and experimental error at different frequencies makes sense. We believe that it is this little difference and thinking that can improve and grow step by step. Regarding the high frequency part, it is not the focus of this article, because the model is also suitable for steady-state. The comparison of tests and models tells us that about 10Hz is the most accurate. Of course, we will also conduct high-frequency related research in the future, hoping to gain something.To make the article more detailed and reliable, we have again made a simple revision:
-Line 173-176:Added description and formula for coil turns.
-Line 177:The source of the current density value range.
-Line 184-186:Added descriptions of the governing equations, meshes, etc.Make manuscript content easily verifiable to confirm its authenticity.
-Line 449-451:Refine the description of the conclusion.
-Line 454 and 455:Unified article description.

Round 3
Reviewer 3 Report
The authors have responded to most of the comments.